# Small-Diameter Subchondral Drilling Improves DNA and Proteoglycan Content of the Cartilaginous Repair Tissue in a Large Animal Model of a Full-Thickness Chondral Defect

**DOI:** 10.3390/jcm9061903

**Published:** 2020-06-18

**Authors:** Patrick Orth, Mona Eldracher, Magali Cucchiarini, Henning Madry

**Affiliations:** Center of Experimental Orthopaedics, Saarland University, Kirrberger Strasse 100, Building 37, D-66421 Homburg, Germany; patrick.orth@uks.eu (P.O.); monaeldracher@gmx.de (M.E.); magali.cucchiarini@uks.eu (M.C.)

**Keywords:** articular cartilage, marrow stimulation, subchondral drilling, early osteoarthritis, extracellular matrix, proteoglycan, Type I collagen, Type II collagen, DNA

## Abstract

This study quantified changes in the DNA content and extracellular matrix composition of both the cartilaginous repair tissue and the adjacent cartilage in a large animal model of a chondral defect treated by subchondral drilling. Content of DNA, proteoglycans, and Type II and Type I collagen, as well as their different ratios were assessed at 6 months in vivo after treatment of full-thickness cartilage defects in the femoral trochlea of adult sheep with six subchondral drill holes, each of either 1.0 mm or 1.8 mm in diameter by biochemical analyses of the repair tissue and the adjacent cartilage and compared with the original cartilage. Only subchondral drilling which were 1.0 mm in diameter significantly increased both DNA and proteoglycan content of the repair tissue compared to the original cartilage. DNA content correlated with the proteoglycan and Type II collagen content within the repair tissue. Significantly higher amounts of Type I collagen within the repair tissue and significantly increased DNA, proteoglycan, and Type I collagen content in the adjacent cartilage were identified. These translational data support the use of small-diameter bone-cutting devices for marrow stimulation. Signs of early degeneration were present within the cartilaginous repair tissue and the adjacent cartilage.

## 1. Introduction

Marrow stimulation techniques, such as subchondral drilling [1], are established first-line treatment options for symptomatic small articular cartilage defects of less than 3 cm^2^ [2,3]. Its common principle is the surgical creation of multiple perforations of the subchondral bone plate, such as by the use of a drill bit, allowing for the access of reparative multipotent progenitor cells from the subchondral bone marrow cavity to the cartilage lesion [4]. Marrow stimulation techniques have been proven to provide good clinical function and pain relief at the short- and mid-term [5]. Long-term outcomes are mainly deteriorated by the incapacity of the cartilaginous repair tissue to permanently withstand mechanical load and prevent perifocal osteoarthritis (OA) [6]. Here, the early phases of this degenerative process have especially been gaining attention as of late; that is, the period of the disease when some regenerative capacity of the articular cartilage is still preserved [7].

Interestingly, the effect of the subchondral drill hole diameter has only recently been investigated, suggesting a supremacy of small- (≤1.0 mm) versus large-diameter (≥1.5 mm) perforations for the histological aspect of the cartilaginous repair tissue and for micro-computed tomography (micro-CT) parameters of subchondral bone reconstitution [8,9]. However, such a possible effect of drill hole size on the extracellular matrix (ECM) composition and molecular properties of the repair tissue remains unknown. Besides, the biochemical composition of the repair cartilage has been compared only very infrequently with those of the original articular cartilage and the cartilage adjacent to the defects [10,11], and almost never in the context of marrow stimulation. Such biochemical evaluations usually rely on an established and standardized determination of DNA (as a surrogate of the cell number), proteoglycan, Type II and Type I collagen content [12,13], and have already been applied successfully to determine the effect of cell-based treatments for articular cartilage defects in large animal models [14,15,16]. Further, the biochemical composition of articular cartilage is thought to be a surrogate for its mechanical properties [15,17].

The present study compares the effect of two different drill hole diameters in a large animal model of the repair of a full-thickness chondral defect on the complex interrelationship of the cells and their ECM within both the cartilaginous repair tissue and the articular cartilage adjacent to the defects at 6 months in vivo. Specifically, we tested the hypothesis that small-diameter drill holes would improve the ECM composition of the cartilaginous repair tissue compared with larger holes. We further analysed the adjacent articular cartilage for signs of early degeneration.

## 2. Methods

### 2.1. Study Design

The aim of this study was to quantify and to correlate the content of DNA and key components of the ECM based on samples obtained in a previous investigation [9] by applying either 1.0 mm or 1.8 mm diameter subchondral drill holes to 4 × 8 mm standardized rectangular full-thickness chondral removal sites (also termed cartilage defects) in the lateral femoral trochlea of adult Merino sheep. The original articular cartilage (also termed normal cartilage) removed at the site of the defect creation was stored. Cartilage removal sites were treated by subchondral drilling, applying two different hole diameters (1.0 and 1.8 mm). After 6 months, animals were sacrificed and the cartilage repair tissues, as well as cartilage samples adjacent to the defects were retrieved. First, the biochemical composition of the repair tissues was compared between defects treated by 1.0 mm and 1.8 mm drill holes (termed 1.0 and 1.8 mm defects). Second, cartilage repair tissues from both treatment groups were compared with normal and adjacent cartilage. Third, biochemical findings were correlated with previously reported data [9] on histological cartilage and microstructural subchondral bone repair.

### 2.2. Animal Experiments

All animal experiments were conducted in agreement with the national legislation on protection of animals and the National Institutes of Health (NIH) Guidelines for the Care and Use of Laboratory Animals (NIH Publication 85-23, Rev 1985) and were approved by the local governmental animal care committee, as previously described [9]. The number of animals needed to treat was calculated based on data from other translational cartilage defect models and according to previous recommendations [18]. Skeletally mature, healthy, female Merino sheep (*n* = 13; average age 35 ± 10 months; average body weight 70 ± 16 kg) received water *ad libitum* and were fed a standard diet. All animals were continuously monitored by a veterinarian. Osteoarthritis was excluded on preoperative radiographs. Anesthesia, surgery, and postoperative treatment were performed as previously described [9,19,20]. Standardized rectangular (4 × 8 mm) full-thickness chondral removal sites were created unilaterally on the lateral facet of the femoral trochlea. The original articular cartilage from the defects was retrieved, stored at −80 °C, and served as the normal control. Six uniform subchondral drill holes (diameters: 1.0 [*n* = 7] or 1.8 mm [*n* = 6]) were then introduced perpendicular to the joint surface within each removal site in a standardized fashion (depth 10.0 mm) using fluted K-wires with a threaded trocar tip under constant irrigation by saline to avoid coagulation or thermal necrosis. The animals were allowed immediate full weightbearing. Animals were sacrificed 6 months post-operatively. The repair tissue of the distal halves of the defects (4 × 4 mm), as well as biopsies of adjacent articular cartilage were collected for further biochemical analyses. Adjacent cartilage samples were retrieved from an area of 4 × 3 mm directly neighboring the defect sites distally. Normal, repair, and adjacent cartilage samples were cautiously retrieved using a scalpel with a #15 blade and Dumont straight forceps with a fine tip. The effect of a subchondral drill hole diameter on histological and micro-CT parameters of osteochondral repair in these 13 animals had already been reported elsewhere [9].

### 2.3. Biochemical Analyses

Biochemical analysis was performed for (1) original cartilage removed during defect creation, (2) repair cartilage of the distal defect halves, and (3) adjacent cartilage retrieved distally to the defects at sacrifice. Thus, 39 samples of 13 animals were digested overnight at 60 °C in a 500 μg/mL papain solution [21,22,23]. DNA content was determined by Hoechst 33258 assay [22]. The bicinchoninic acid test was used for detecting general protein contents. Proteoglycan concentrations were measured spectrophotometrically by binding to dimethylmethylene blue (DMMB) [23,24], with chondroitin-6-sulfate serving as the standard. Type I and Type II collagen content was determined using enzyme-linked immunosorbent assays (ELISA; MD Bioproducts, Saint Paul, MN, USA). All measurements were performed using a spectrophotometer/fluorometer (GENios, Tecan, Crailsheim, Germany).

### 2.4. Statistical Analysis

The two-sample t-test with unequal variances for independent, non-parametric data was applied to compare results between 1.0 mm and 1.8 mm defects. For within-group comparisons, the Wilcoxon signed-rank test for dependent, non-parametric data was employed. To determine the strength of association between the obtained biochemical data and (1) the histological average total score value of the repair tissue according to Sellers et al. [25], as well as (2) major micro-CT parameters of subchondral bone reconstitution (bone mineral density [BMD] and bone volume fraction [BV/TV]) as reported earlier for these defects [9], the non-parametric Spearman correlation coefficient (r) was used. A two-tailed value of *p* < 0.05 was considered significant. All calculations were performed with OriginPro 8G (OriginLab Corporation, Northampton, MA, USA).

## 3. Results

### 3.1. Biochemical Evaluation of the Original Articular Cartilage

DNA, proteoglycan, Type I and Type II collagen content of the normal articular cartilage, removed intraoperatively during the defect creation in vivo, as well as the proteoglycan/DNA, Type II collagen/DNA, Type I collagen/DNA, and Type I/Type II collagen ratios were not significantly different between the two groups with the different drill hole diameters (Table 1).

Biochemical evaluation of the articular cartilage repair tissue:

Upon evaluation of the cartilaginous repair tissue retrieved at 6 months postoperatively, the DNA, proteoglycan, Type I and Type II collagen contents, as well as the proteoglycan/DNA, Type II collagen/DNA, Type I collagen/DNA, and Type I/Type II collagen ratios did not vary significantly between the two groups (Table 1). DNA and proteoglycan content was 1.4- and 1.7-fold higher in 1.0 mm diameter defects compared with 1.8 mm diameter defects, without reaching statistical significance (Table 1; *p* = 0.344 and *p* = 0.162, respectively). Type II collagen content in the 1.0 mm diameter defects was 2.4-fold higher, together with a 1.5-fold higher Type II collagen/DNA ratio compared to 1.8 mm diameter defects (both *p* ≥ 0.215). Type I collagen also increased 2.3-fold in 1.0 mm diameter defects, and the Type I collagen/DNA ratio was 1.9-fold higher (both *p* ≥ 0.127) (Table 1).

When compared to the normal cartilage retrieved at the site of the defect, DNA (3.9-fold) and proteoglycan (4.2-fold) content was significantly increased in 1.0 mm diameter defects (both *p* ≤ 0.015), but not following 1.8 mm drilling (*p* = 0.072) (Table 2, Figure 1A–D). The repair tissue of both groups also contained significantly more Type I collagen (60.8-fold) and had a higher Type I collagen/DNA ratio (20.0-fold) than the original cartilage (1.0 mm defects: *p* ≤ 0.022; 1.8 mm defects: *p* ≤ 0.016) (Table 1, Figure 2G–J, Figure 3).

### 3.2. Biochemical Analysis of the Adjacent Articular Cartilage

DNA, proteoglycan, Type I and Type II collagen content of the cartilage adjacent to the treated cartilage defects, as well as the proteoglycan/DNA, Type II collagen/DNA, Type I collagen/DNA, and Type I/Type II collagen ratios were not different between the two treatment groups (Table 1).

Compared to the normal articular cartilage, the cartilage adjacent to the defect treated with the two different drill hole diameters contained significantly increased DNA (1.0 mm defects: 2.6-fold, *p* < 0.001; 1.8 mm defects: 2.0-fold, *p* = 0.006), proteoglycan (1.0 mm defects: 2.2-fold, *p* < 0.001; 1.8 mm defects: 1.8-fold, *p* = 0.037), and Type I collagen content (1.0 mm defects: 47.9-fold, *p* = 0.011; 1.8 mm defects: 55.5-fold, *p* = 0.012) and Type I collagen/DNA ratios (1.0 mm defects: 18.3-fold, *p* = 0.008; 1.8 mm defects: 23.0-fold, *p* = 0.011) (Table 1, Table 2, Figure 3). No significant differences in Type II collagen content (1.0 mm defects: 1.3-fold, *p* = 0.399; 1.8 mm defects: 1.7-fold, *p* = 0.063) and proteoglycan/DNA ratios (1.0 mm defects: 0.9-fold, *p* = 0.252; 1.8 mm defects: 0.9-fold, *p* = 0.562) existed between adjacent and original articular cartilage (Table 2, Figure 3).

### 3.3. Correlation Analysis between the Biochemical Parameters and Structural Indices of Osteochondral Repair

In both treatment groups, DNA content of the repair tissue correlated with the proteoglycan content (*r* ≥ 0.829; *p* ≤ 0.042) and with its Type II collagen content (*r* ≥ 0.714; *p* ≤ 0.005). The Type II collagen content also correlated with the proteoglycan content (*r* ≥ 0.943; *p* ≤ 0.005). Other correlations were beyond statistical significance.

Within the articular cartilage adjacent to the treated chondral defects, Type II collagen content also correlated with the proteoglycan content (*r* ≥ 0.821; *p* ≤ 0.023). No other significant correlations existed between DNA, proteoglycan, and collagen content.

No significant correlations were detected between all biochemical parameters of the repair tissue reported here and the historical histological average total score value of the repair tissue [25] (1.0 mm defects: 19.21 ± 4.80, 0.011 < *r* < 0.751, 0.051 < *p* < 0.964; 1.8 mm defects: 25.43 ± 3.73, 0.085 < *r* < 0.544, 0.265 < *p* < 0.873). Individual histological score values, as well as the historical major micro-CT parameters of the reconstituted subchondral bone plate and subarticular spongiosa (BMD: 0.085 < *r* < 0.772, 0.071 < *p* < 0.873; BV/TV: 0.028 < *r* < 0.773, 0.069 < *p* < 0.958) [9] did not significantly correlate.

## 4. Discussion

The present study quantified the effect of two different drill hole diameters on the cells and their ECM within the cartilaginous repair tissue and the articular cartilage adjacent to the defects in a large animal model of marrow-stimulation-based chondral repair at 6 months in vivo. There are a number of important conclusions arising from these data that expand our insight into this complex interrelationship during the repair of focal articular cartilage defects and potential progression to joint degeneration. The first major finding is that only 1.0 mm diameter subchondral drilling significantly increases mean DNA and proteoglycan content of the cartilaginous repair tissue compared to the original cartilage, but not 1.8 mm drilling, supporting the value of small-diameter bone-cutting devices for marrow stimulation. Second, DNA content correlated with the proteoglycan and Type II collagen content (which also correlated among themselves), suggesting an orchestrated deposition of ECM within the cartilaginous repair tissue following subchondral drilling. Finally, and perhaps surprisingly, signs of early tissue degeneration were already present both within the (fibro)cartilaginous repair tissue, as indicated by the higher biochemical indices for Type I collagen, and the articular cartilage adjacent to the defects, as revealed by the significantly increased DNA, proteoglycan, and Type I collagen content without differences between the two treatment groups at 6 months in vivo.

Accurately assessing cells and ECM content within the cartilaginous repair tissue and the adjacent articular cartilage by means of biochemical evaluations is mandatory for any comprehensive assessment of the repair process, as it reflects the extent to which biological activity is stimulated in the cells, forming the repair tissue and its vicinity [12]. These activities include cell number and proliferation—reflective of the repopulation of a defect—as well as biosynthesis of ECM components required to support the cartilage architecture and organization. Analysis of cell number and proliferation is regularly performed by estimating DNA concentrations, such as by determining intercalation of a fluorescent DNA dye [22]. Determination of ECM biosynthesis generally relies on the evaluation of proteoglycan and collagen concentrations by DMMB assay or ELISA, respectively [12,26]. Such methods are established to monitor the extent of cartilage repair using human samples [27] and in various animal models, including rats [28], rabbits [29,30], sheep or goats [31,32], and horses [33,34], substantiating their feasibility in clinically relevant settings. Furthermore, proteoglycan and collagen content has been proven to mainly account for the compressive properties of normal articular cartilage [17,35], underlining the importance of these parameters for cartilage repair strategies. To date, biomechanical testing of the articular cartilage repair tissue following subchondral drilling has never been performed in sheep, and only once in goats [36], underlining the need for further biomechanical testing in translational investigations.

The data revealed considerable higher amounts of DNA (up to 3.9-fold), proteoglycan (up to 4.2-fold), and Type II collagen (up to 4.0-fold) content in the cartilaginous repair tissue following subchondral drilling. Of note, following small (1.0 mm) diameter drilling, these numerical differences were significant for mean DNA and proteoglycan concentrations (albeit not all individual animals always exhibited such significant improvement). Yet, as this effect was not observed following large (1.8 mm) diameter drilling, the data suggest a beneficial effect of small drill holes on the cell and ECM content of the repair tissue. Interestingly, this finding is in good agreement with the histological assessment of the cartilaginous repair tissue and the micro-CT analysis of the subchondral bone in these defects [9]. Furthermore, regarding the clinically more frequently applied microfracture technique [37], a recent investigation confirmed that small-diameter awls improve articular cartilage repair in sheep more effectively than larger awls [8]. Of note, the ECM might further strengthen over time, as clinical studies suggest a maturation of the repair tissue over several years [38,39].

Compared to the normal cartilage, Type I collagen content was significantly increased up to 60-fold within the repair tissue of either treatment group. In their classical biochemical studies, Glimcher and coworkers identified Type II collagen as only predominant at 1–2 months in a rabbit defect model, while Type I collagen persisted for up to 1 year [10]. Thus, the benefits of increased proteoglycan and DNA content in the repair tissue following subchondral drilling at 6 months postoperatively need to be weighed against the significant presence of Type I collagen, which is associated with a fibrocartilaginous phenotype. This finding is supported by the reported occurrence of moderate to severe surface fibrillation and irregularity of the repair tissue [9] which corresponds to an OARSI (Osteoarthritis Research Society International) grade 3.0 [40], thus unveiling fibrocartilaginous characteristics [41]. While articular cartilage repair is often regarded as the sole important outcome of studies on focal cartilage defects, it is important to remember that even small chondral lesions may induce early OA in the affected joint compartment [42], as shown in a similar model by Schinhan and Nehrer [6] and in long-term clinical evaluations [43,44].

OA is an insidious disease, and may be present long before arising to a clinically symptomatic state. We substantiated degenerative changes within the adjacent articular cartilage following either subchondral drilling technique, exhibiting an up to 55-fold increase in Type I collagen content compared with the normal cartilage. Interestingly, no significant differences in Type II collagen content was identified, indicating that the collagenous network of the adjacent cartilage was not yet affected. Both DNA and proteoglycan content was significantly elevated, most likely representing an early attempt to restore these lost ECM components adjacent to the lesion, as seen in the early hypertrophic phase of OA, suggesting that at this time-point, the rate of repair is greater than the rate of degradation [45]. Altogether, these findings point towards an early, but not advanced stage of OA within the articular cartilage adjacent to the treated cartilage removal sites [7]. However, the presented data do not allow to judge whether subchondral drilling per se has the potential to diminish degenerative changes within the repaired and adjacent cartilage. For this purpose, a control group of untreated cartilage removal sites would be necessary. Although marrow stimulation could not prevent such degenerative changes in a sheep study applying the microfracture technique [8], this important aspect remains to be elucidated in future investigation of subchondral drilling.

Of note, proteoglycan content of the cartilaginous repair tissue correlated with DNA and Type II collagen content in both groups, which is in good agreement with previous studies [45,46]. Interestingly, no significant correlation was found between the evaluated biochemical parameters of the cartilaginous repair tissue and its histological or radiological parameters of osteochondral repair reported earlier for these defects [9]. The absence of such external correlations between different methods of evaluation is in line with preceding reports, indicating that time-dependent osteochondral repair mechanisms are lacking synchronization [47] and proceed at a different pace [48] in both small and large animal models [9]. Thus, biochemical, histological, and radiological evaluations may be considered as complementary, rather than replaceable tools to assess experimental articular cartilage repair in vivo [46]. It remains to be elucidated whether these parameters may correlate with functional results of patients in a clinical situation.

Possible limitations of this study include the selection of a single time-point postoperatively, precluding an assessment of changes over time, as stipulated by our animal license, and the lack of biomechanical testing. However, the possibility of comparing for each animal the characteristics of the original tissue present at the site and the time of articular cartilage removal with its individual repair tissue at the identical location allowed for important insights, besides the comparison between the 1.0 and 1.8 mm diameter instruments. Such comparison with the normal control in a pre- versus post-treatment longitudinal study design is an established and reliable statistical approach [49,50,51]. Other strengths are the inclusion of several relevant and objective biochemical parameters, as well as their correlation with a number of indices of osteochondral repair, serving as historical control and a gold standard of evaluation. With a view on OA possibly encroaching on formerly unaffected joint areas originating from such defects, it will be important to complement these biochemical observations at mid-term with data from longer observation periods to answer the question on whether the reported changes may further progress towards higher degrees of OA in the future.

## 5. Conclusions

This study provides insight into the complex interrelationship of cells and their ECM during the repair of focal articular cartilage defects and their potential progression to joint degeneration between a focal chondral defect in a previously normal knee joint during its repair, based on a marrow-stimulation technique and the affected adjacent articular cartilage. From a clinical perspective, these translational data support the hypothesis that small-diameter bone-cutting devices improve marrow-stimulation of a focal chondral defect, although signs of early degeneration were present not only within the cartilaginous repair tissue, but also in the adjacent articular cartilage.

## Figures and Tables

**Figure 1 jcm-09-01903-f001:**
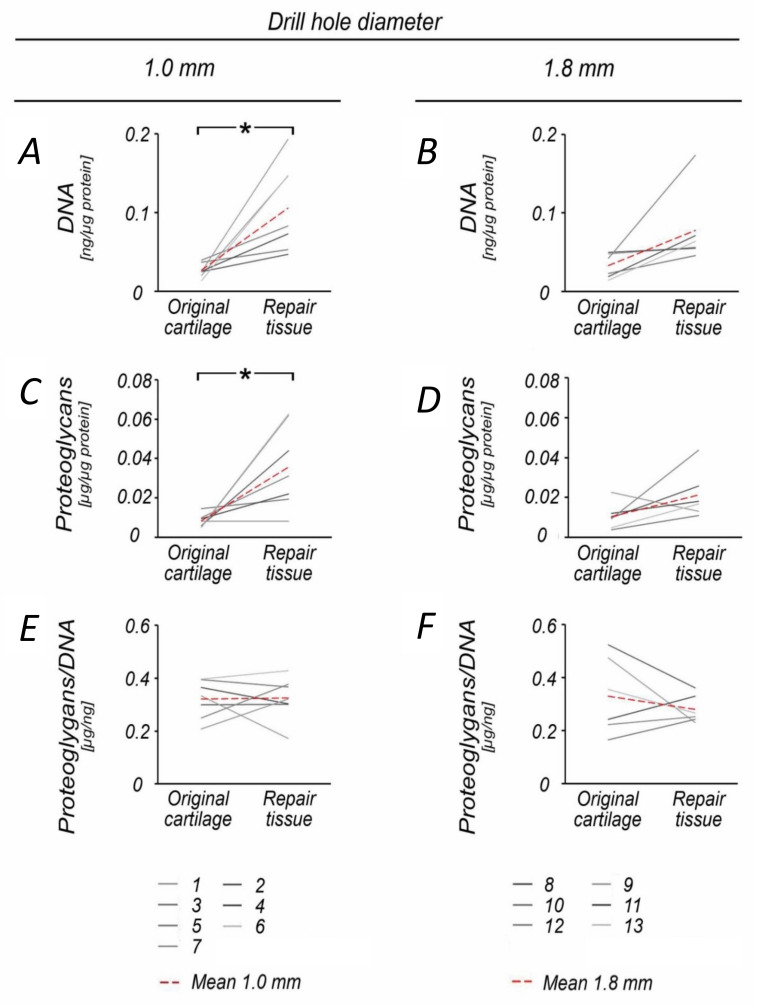
Comparison of proteoglycan and DNA content within original and repair cartilage following 1.0 and 1.8 mm subchondral drilling. Following 1.0 mm drilling, repair cartilage exhibited significantly increased DNA (**A**) and proteoglycan (**C**) content when compared with the original cartilage. In contrast, DNA and proteoglycan content was not significantly affected in the group with 1.8 mm drilling (**B**,**D**). The proteoglycan/DNA ratio was balanced in the 1.0 mm group, (**E**) whereas it decreased in the 1.8 mm group (**F**). Numbers 1–13 represent individual animals. Red dotted lines indicate mean values. * *p* < 0.05.

**Figure 2 jcm-09-01903-f002:**
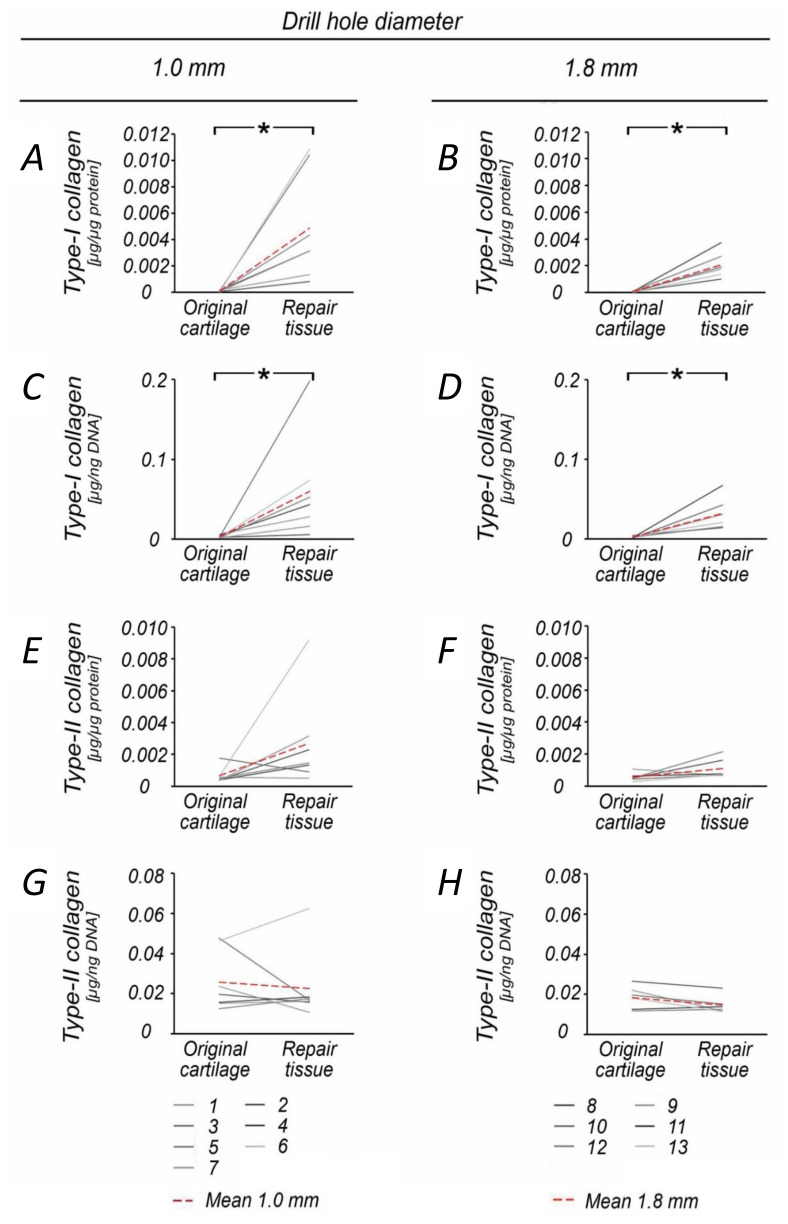
Comparison of Type I and Type II collagen content within original and repair cartilage following 1.0 and 1.8 mm subchondral drilling. Within both groups, the repair cartilage showed significantly increased Type I collagen content (**A**,**B**) and Type I collagen/DNA ratio (**C**,**D**) when compared to the original articular cartilage, reflective of early osteoarthritis. Analysis of Type II collagen content revealed no significant differences between repair and original cartilage (**E**–**H**). Numbers 1–13 represent individual animals. Red dotted lines indicate mean values. * *p* < 0.05.

**Figure 3 jcm-09-01903-f003:**
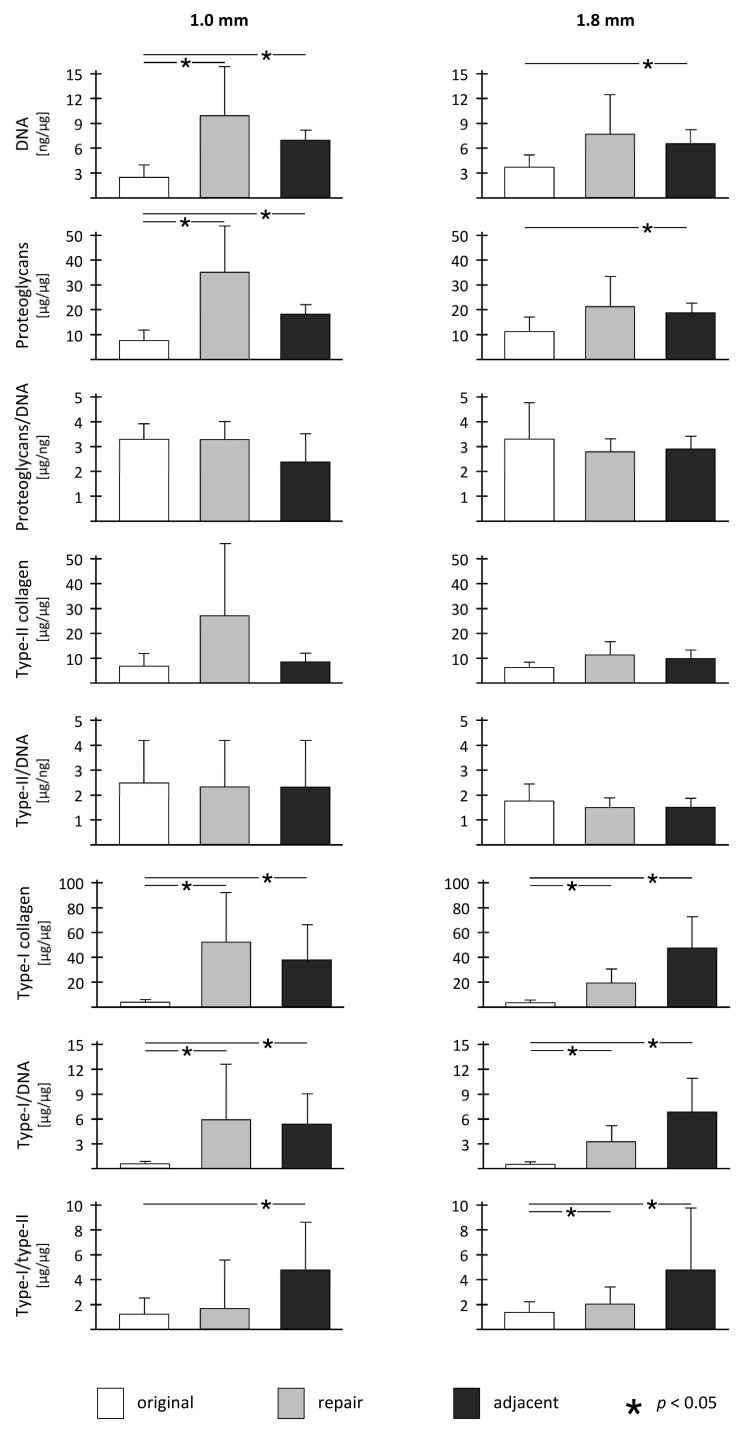
Overview of biochemical parameters and statistical comparison between original cartilage, repair cartilage, and adjacent cartilage within both treatment groups.

**Table 1 jcm-09-01903-t001:** Overview of the biochemical parameters obtained within original cartilage, repair cartilage, and adjacent cartilage.

Specimen		Unit	1.0 mm	1.8 mm	x-Fold Difference	*p*
Original cartilage	DNA	[ng/μg]	2.7^−2^ ± 0.9^−2^	3.3^−2^ ± 1.6^−2^	1.2	0.435
	Proteoglycans	[μg/μg]	8.4^−3^ ± 3.3^−3^	10.5^−3^ ± 6.7^−3^	1.3	0.514
	Proteoglycans/DNA	[μg/ng]	3.2^−1^ ± 0.7^−1^	3.3^−1^ ± 1.5^−1^	1.0	0.878
	Type II collagen	[μg/μg]	6.7^−4^ ± 5.0^−4^	5.6^−4^ ± 2.7^−4^	1.2	0.644
	Type II collagen/DNA	[μg/ng]	2.6^−2^ ± 1.5^−2^	1.8^−2^ ± 0.6^−2^	1.4	0.261
	Type I collagen	[μg/μg]	0.8^−4^ ± 0.5^−4^	0.8^−4^ ± 0.3^−4^	1.1	0.741
	Type I collagen/DNA	[μg/ng]	0.3^−2^ ± 0.2^−2^	0.3^−2^ ± 0.1^−2^	1.2	0.455
	Type I/Type II collagen	[μg/μg]	1.3^−1^ ± 1.0^−1^	1.4^−1^ ± 0.7^−1^	0.9	0.688
Repair cartilage	DNA	[ng/μg]	10.5^−2^ ± 5.6^−2^	7.7^−2^ ± 4.8^−2^	1.4	0.344
	Proteoglycans	[μg/μg]	35.6^−3^ ± 21.3^−3^	21.4^−3^ ± 12.1^−3^	1.7	0.162
	Proteoglycans/DNA	[μg/ng]	3.2^−1^ ± 0.8^−1^	2.8^−1^ ± 0.5^−1^	1.2	0.271
	Type II collagen	[μg/μg]	26.9^−4^ ± 30.0^−4^	11.0^−4^ ± 6.4^−4^	2.4	0.215
	Type II collagen/DNA	[μg/ng]	2.3^−2^ ± 1.8^−2^	1.5^−2^ ± 0.4^−2^	1.5	0.287
	Type I collagen	[μg/μg]	48.6^−4^ ± 41.2^−4^	20.8^−4^ ± 9.9^−4^	2.3	0.127
	Type I collagen/DNA	[μg/ng]	6.0^−2^ ± 6.5^−2^	3.2^−2^ ± 2.0^−2^	1.9	0.324
	Type I/Type II collagen	[μg/μg]	1.8 ± 3.9	1.9 ± 1.5	1.0	0.505
Adjacent cartilage	DNA	[ng/μg]	6.9^−2^ ± 1.2^−2^	6.6^−2^ ± 1.7^−2^	1.1	0.679
	Proteoglycans	[μg/μg]	18.7^−3^ ± 3.3^−3^	18.6^−3^ ± 4.2^−3^	1.0	0.968
	Proteoglycans/DNA	[μg/ng]	2.8^−1^ ± 0.6^−1^	2.9^−1^ ± 0.6^−1^	1.1	0.668
	Type II collagen	[μg/μg]	8.7^−4^ ± 3.5^−4^	9.3^−4^ ± 3.4^−4^	1.1	0.752
	Type II collagen/DNA	[μg/ng]	2.3^−2^ ± 1.8^−2^	1.5^−2^ ± 0.4^−2^	1.5	0.287
	Type I collagen	[μg/μg]	38.3^−4^ ± 27.3^−4^	44.4^−4^ ± 28.0^−4^	1.2	0.701
	Type I collagen/DNA	[μg/ng]	5.5^−2^ ± 3.5^−2^	6.9^−2^ ± 4.2^−2^	1.3	0.532
	Type I/Type II collagen	[μg/μg]	4.4 ± 4.1	4.8 ± 4.0	0.9	0.787

**Table 2 jcm-09-01903-t002:** Statistical comparison between the biochemical parameters obtained within original cartilage, repair cartilage, and adjacent cartilage.

					*p*					
Drill Hole Diameter	Specimen		DNA	Proteoglycans	Proteoglycans/DNA	Type II Collagen	Type II Collagen/DNA	Type I Collagen	Type I Collagen/DNA	Type I/ Type II Collagen
			[ng/μg]	[μg/μg]	[μg/ng]	[μg/μg]	[μg/ng]	[μg/μg]	[μg/ng]	[μg/μg]
1.0 mm	Original cartilage	Repair cartilage	**0.010**	**0.015**	0.935	0.128	0.722	**0.022**	**0.020**	0.062
	Original cartilage	Adjacent cartilage	**<0.001**	**<0.001**	0.252	0.399	0.064	**0.011**	**0.008**	**0.011**
	Repair cartilage	Adjacent cartilage	0.136	0.082	0.250	0.161	0.208	0.595	0.877	0.447
1.8 mm	Original cartilage	Repair cartilage	0.072	0.089	0.452	0.096	0.223	**<0.001**	**0.016**	**0.005**
	Original cartilage	Adjacent cartilage	**0.006**	**0.037**	0.562	0.063	0.196	**0.012**	**0.011**	**0.011**
	Repair cartilage	Adjacent cartilage	0.596	0.610	0.737	0.565	0.920	0.099	0.092	0.133

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
