# Peer review of "Small-Diameter Subchondral Drilling Improves DNA and Proteoglycan Content of the Cartilaginous Repair Tissue in a Large Animal Model of a Full-Thickness Chondral Defect"

_jcm, 2020, doi:10.3390/jcm9061903_

Round 1

Reviewer 1 Report

The authors have addressed all my concerns/questions. Thank you.

Reviewer 2 Report

No further comments or remarks.

This manuscript is a resubmission of an earlier submission. The following is a list of the peer review reports and author responses from that submission.

Round 1

Reviewer 1 Report

Dear Authors,

I have had the pleasure to review your work evaluating the histological characteristics of the newly formed tissue after treatment of chondral defects with different drilling hole techniques.The investigation is well done and the paper easy to read and understandable for the casual reader. The message provided is also clear. Some flaws or limitations of the current work included the only one time point postoperatively assessment, making it impossible to verify if there changes over time, as well as the lack of biomechanical assessment of the repaired area. So we have to assume that a better DNA content and ECM mean a better newly formed cartilage. 

Title

It is clear and self-explanative.

Abstract

Correct according to the core of the manuscript

Introduction

Nice background of the marrow stimulation techniques. Both purpose and hypothesis are correctly stated in the last paragraph of this section.

Methods

Overall, this section is clear and well described. However, I have some questions and remarks.

With regards to the Animal experiments. Not only the size of the drilling device but the drilling technique it self might be relevant for the outcomes. Could you describe how was it done? I guess you have used “cold drilling” to avoid coagulation or thermal necrosis.

The behaviour of subchondral bone is an important point when evaluating cartilage repair. Since you already got this data from a previous investigation, was it easy to correlate both figures?

I miss a biomechanical study in the present work. Don’t you think that the biomechanical information would be relevant to the aim of the current study?

Concerning the Biochemical analysis sub-section. What extend of adjacent cartilage was assessed (i.e. 1mm, 2mm, etc)? How this adjacent cartilage was retrieved? I think this is relevant information to understand the real value of the results.

Results

This section is clear. However, it is quite surprising the lack of correlation between histological score values and the historical micro-CT parameters corresponding to the subchondral bone (?)

Discussion

Well structured. No further comments.

Author Response

Point-by-point response to the Reviewers’ comments

We were delighted to receive the review for our manuscript, and appreciate the Reviewers’ thorough reviews and insightful comments and suggestions to improve the quality and impact of our work. We have addressed each of the concerns below, on a point-by-point basis and making changes to the manuscript where appropriate. We believe that the manuscript has been significantly strengthened as a result of these recommendations. We very much appreciate the opportunity to submit this revised manuscript, and do hope that, with these changes and clarifications, our work might be suitable for publication in the Journal of Clinical Medicine.

Thanks again and best regards,

Henning Madry

Reviewer 1

1. Reviewer comment: I have had the pleasure to review your work evaluating the histological characteristics of the newly formed tissue after treatment of chondral defects with different drilling hole techniques. The investigation is well done and the paper easy to read and understandable for the casual reader. The message provided is also clear.

Author response: We thank the Reviewer very much for this remark and overall for the positive and very constructive and helpful review of our manuscript.

Author action: None.

2. Reviewer comment: With regards to the Animal experiments. Not only the size of the drilling device but the drilling technique itself might be relevant for the outcomes. Could you describe how was it done? I guess you have used “cold drilling” to avoid coagulation or thermal necrosis.

Author response: We again thank the Reviewer very much for this helpful remark. Indeed, the surgical technique of subchondral drilling and the applied surgical instruments are of high relevance for the outcome of the procedure. Here, fluted K-wires of the respective diameter with a threaded trocar tip were used in order to establish standardized perforations. Rotating K-wires were constantly irrigated by saline in order to avoid thermal necrosis during the drilling procedure.

Author action: The surgical drilling technique is now explained more in detail in the revised version of the manuscript and more emphasis is laid on the aspect of irrigation in order to avoid coagulation or osteonecrosis (page 7, lines 19-24).

3. Reviewer comment: The behaviour of subchondral bone is an important point when evaluating cartilage repair. Since you already got this data from a previous investigation, was it easy to correlate both figures?

Author response: We again thank the Reviewer very much for this attentive comment. We fully agree that subchondral bone reconstitution following osteochondral injuries or surgical marrows stimulation procedures is underrepresented in the currently available literature. We could recently demonstrate in a systematic literature search among PubMed, Scopus and ScienceDirect [1] that subchondral bone reconstruction following such drilling treatment was only assessed in very few animal studies to date, using either semiquantitative scoring [2], histomorphometric quantification of subchondral bone alterations [3] or quantitative assessments using micro-CT [2,4,5].

As stated in the manuscript, the historical data of subchondral bone analysis by micro-CT as well as of articular cartilage repair by histology were previously reported in our manuscript by Eldracher et al. [5]. To determine the strength of association between the obtained biochemical data and major micro-CT parameters of subchondral bone reconstitution (bone mineral density [BMD] and bone volume fraction [BV/TV]), the non-parametric Spearman correlation coefficient (r) was used (as explained in the previously submitted version of the manuscript). Such correlation analysis is laborious but facilitated when only selected key micro-CT parameters are included. Therefore, we concentrated in the present manuscript on BMD and BV/TV.

The results showed no significant correlations between all biochemical parameters of the repair tissue and these micro-CT parameters of the reconstituted subchondral bone plate as well as the subarticular spongiosa. Further, biochemical parameters did not correlate with the histological average total score value and all individual histological subscore items.

This finding is not surprising but confirms the results of our previous work [6,7], indicating that the subchondral bone compartment is reconstituted in a distinct chronological order but lacks correlation with the histological aspect of articular cartilage repair; thus subchondral bone and articular cartilage repair proceed at a different pace [3,8].

Author action: In the revised version of the manuscript, we now put more emphasis on the fact that the processes of articular cartilage repair and subchondral bone reconstitution lack synchronization during osteochondral repair. This highly interesting aspect is now discussed more in detail (page 26, lines 15-23).

4. Reviewer comment: I miss a biomechanical study in the present work. Don’t you think that the biomechanical information would be relevant to the aim of the current study?

Author response: We thank the Reviewer again very much for this suggestion and fully agree on the utmost importance of biomechanical testing of articular cartilage. To date - underlining the scientific value of this Reviewer’s comment - biomechanical testing of the articular cartilage repair tissue following subchondral drilling has never been performed in sheep and only once in goats [9]. Yet, the here applied methodology of biochemical assessment of previously obtained osteochondral specimen of sheep stifle joints does not allow for any further biomechanical testing of these samples due to their prior destructive processing for histological analysis.    

Author action: In the Discussion section, we acknowledge the lack of biomechanical testing as a limitation of the study and further stress on the importance of the biomechanical assessment of articular cartilage repair tissue (page 27, lines 1-3). Unfortunately, we are unable to add such data to the revised version of the manuscript for the reasons given above. The reference of Lind and co-workers [9], the only available study to date that has published results of biomechanical testing of articular cartilage repair tissue following subchondral drilling in a large animal model, has been amended; the reference list has been renumbered accordingly (page 33, reference #37).    

5. Reviewer comment: Concerning the Biochemical analysis sub-section. What extend of adjacent cartilage was assessed (i.e. 1mm, 2mm, etc)? How this adjacent cartilage was retrieved? I think this is relevant information to understand the real value of the results.

Author response: We thank the Reviewer once again for this attentive remark and recognize that the technique of specimen retrieval was unfortunately not sufficiently described in detail in the initially submitted version of the manuscript. Adjacent cartilage samples were retrieved from an area of 4 mm in width and 3 mm in length, directly neighboring the defect sites distally. Normal, repair and adjacent cartilage samples were cautiously retrieved using a scalpel with a blade #15 and Dumont straight forceps with a fine tip.

Author action: We now explain more in detail the exact location for the retrieval of adjacent cartilage specimen as well as the respective surgical technique and applied instruments (page 8, lines 2-5).

6. Reviewer comment: Results: This section is clear. However, it is quite surprising the lack of correlation between histological score values and the historical micro-CT parameters corresponding to the subchondral bone (?)

Author response: We thank the Reviewer very much for this important comment. Indeed, and as stated above (please see comment #3), the lack of correlation between histological analysis of articular cartilage repair and radiological (micro-CT) findings concerning subchondral bone reconstitution is interesting and has already been reported earlier by our group for the repair of osteochondral defects in rabbits[7] and following subchondral drilling of chondral defects in sheep [5]. We interpret these findings in such a way that improved subchondral bone repair is not the cause for the simultaneously observed enhanced articular cartilage repair, underscoring the need for more studies to unravel these complex mechanisms of osteochondral repair.

Author action: According to comment #3 of this Reviewer, we now have put more emphasis on the fact that the processes of articular cartilage repair and subchondral bone reconstitution lack synchronization during osteochondral repair. Further, we discuss that it remains to be elucidated whether these parameters may affect the functional outcome following marrow stimulation procedures in patients. We have amended the Discussion to include this important issue (page 26, lines 15-23).

References

  1. Gao, L.; Goebel, L.K.H.; Orth, P.; Cucchiarini, M.; Madry, H. Subchondral drilling for articular cartilage repair: a systematic review of translational research. Dis Model Mech 2018, 11.
  2. Chen, H.; Chevrier, A.; Hoemann, C.D.; Sun, J.; Lascau-Coman, V.; Buschmann, M.D. Bone marrow stimulation induces greater chondrogenesis in trochlear vs condylar cartilage defects in skeletally mature rabbits. Osteoarthritis Cartilage 2013, 21, 999-1007.
  3. Orth, P.; Cucchiarini, M.; Kaul, G.; Ong, M.F.; Graber, S.; Kohn, D.M.; Madry, H. Temporal and spatial migration pattern of the subchondral bone plate in a rabbit osteochondral defect model. Osteoarthritis Cartilage 2012, 20, 1161-1169.
  4. Chen, H.; Chevrier, A.; Hoemann, C.D.; Sun, J.; Ouyang, W.; Buschmann, M.D. Characterization of subchondral bone repair for marrow-stimulated chondral defects and its relationship to articular cartilage resurfacing. Am J Sports Med 2011, 39, 1731-1740.
  5. Eldracher, M.; Orth, P.; Cucchiarini, M.; Pape, D.; Madry, H. Small subchondral drill holes improve marrow stimulation of articular cartilage defects. Am J Sports Med 2014, 42, 2741-2750.
  6. Orth, P.; Cucchiarini, M.; Wagenpfeil, S.; Menger, M.D.; Madry, H. PTH [1-34]-induced alterations of the subchondral bone provoke early osteoarthritis. Osteoarthritis Cartilage 2014, 22, 813-821.
  7. Orth, P.; Cucchiarini, M.; Zurakowski, D.; Menger, M.D.; Kohn, D.M.; Madry, H. Parathyroid hormone [1-34] improves articular cartilage surface architecture and integration and subchondral bone reconstitution in osteochondral defects in vivo. Osteoarthritis Cartilage 2013, 21, 614-624.
  8. Orth, P.; Cucchiarini, M.; Kohn, D.; Madry, H. Alterations of the subchondral bone in osteochondral repair--translational data and clinical evidence. Eur Cell Mater 2013, 25, 299-316.
  9. Lind, M.; Larsen, A.; Clausen, C.; Osther, K.; Everland, H. Cartilage repair with chondrocytes in fibrin hydrogel and MPEG polylactide scaffold: an in vivo study in goats. Knee Surg Sports Traumatol Arthrosc 2008, 16, 690-698.
  10. Goebel, L.; Orth, P.; Muller, A.; Zurakowski, D.; Bucker, A.; Cucchiarini, M.; Pape, D.; Madry, H. Experimental scoring systems for macroscopic articular cartilage repair correlate with the MOCART score assessed by a high-field MRI at 9.4 Tesla - comparative evaluation of five macroscopic scoring systems in a large animal cartilage defect model. Osteoarthritis Cartilage 2012, 20, 1046-1055.
  11. Goebel, L.; Zurakowski, D.; Muller, A.; Pape, D.; Cucchiarini, M.; Madry, H. 2D and 3D MOCART scoring systems assessed by 9.4 T high-field MRI correlate with elementary and complex histological scoring systems in a translational model of osteochondral repair. Osteoarthritis Cartilage 2014, 22, 1386-1395.
  12. Orth, P.; Duffner, J.; Zurakowski, D.; Cucchiarini, M.; Madry, H. Small-Diameter Awls Improve Articular Cartilage Repair After Microfracture Treatment in a Translational Animal Model. Am J Sports Med 2016, 44, 209-219.
  13. Orth, P.; Goebel, L.; Wolfram, U.; Ong, M.F.; Graber, S.; Kohn, D.; Cucchiarini, M.; Ignatius, A.; Pape, D.; Madry, H. Effect of subchondral drilling on the microarchitecture of subchondral bone: analysis in a large animal model at 6 months. Am J Sports Med 2012, 40, 828-836.
  14. Orth, P.; Zurakowski, D.; Alini, M.; Cucchiarini, M.; Madry, H. Reduction of sample size requirements by bilateral versus unilateral research designs in animal models for cartilage tissue engineering. Tissue Eng Part C Methods 2013, 19, 885-891.
  15. Squires, G.R.; Okouneff, S.; Ionescu, M.; Poole, A.R. The pathobiology of focal lesion development in aging human articular cartilage and molecular matrix changes characteristic of osteoarthritis. Arthritis Rheum 2003, 48, 1261-1270.
  16. Madry, H.; Luyten, F.P.; Facchini, A. Biological aspects of early osteoarthritis. Knee Surg Sports Traumatol Arthrosc 2012, 20, 407-422.
  17. Schinhan, M.; Gruber, M.; Vavken, P.; Dorotka, R.; Samouh, L.; Chiari, C.; Gruebl-Barabas, R.; Nehrer, S. Critical-size defect induces unicompartmental osteoarthritis in a stable ovine knee. J Orthop Res 2012, 30, 214-220.
  18. Sanders, T.L.; Pareek, A.; Obey, M.R.; Johnson, N.R.; Carey, J.L.; Stuart, M.J.; Krych, A.J. High Rate of Osteoarthritis After Osteochondritis Dissecans Fragment Excision Compared With Surgical Restoration at a Mean 16-Year Follow-up. Am J Sports Med 2017, 45, 1799-1805.
  19. Gomoll, A.H.; Farr, J.; Gillogly, S.D.; Kercher, J.; Minas, T. Surgical management of articular cartilage defects of the knee. J Bone Joint Surg Am 2010, 92, 2470-2490.
  20. Osterhoff, G.; Loffler, S.; Steinke, H.; Feja, C.; Josten, C.; Hepp, P. Comparative anatomical measurements of osseous structures in the ovine and human knee. Knee 2011, 18, 98-103.
  21. Murshed, K.A.; Cicekcibasi, A.E.; Karabacakoglu, A.; Seker, M.; Ziylan, T. Distal femur morphometry: a gender and bilateral comparative study using magnetic resonance imaging. Surg Radiol Anat 2005, 27, 108-112.
  22. Hamanishi, M.; Nakasa, T.; Kamei, N.; Kazusa, H.; Kamei, G.; Ochi, M. Treatment of cartilage defects by subchondral drilling combined with covering with atelocollagen membrane induces osteogenesis in a rat model. J Orthop Sci 2013, 18, 627-635.
  23. Menche, D.S.; Frenkel, S.R.; Blair, B.; Watnik, N.F.; Toolan, B.C.; Yaghoubian, R.S.; Pitman, M.I. A comparison of abrasion burr arthroplasty and subchondral drilling in the treatment of full-thickness cartilage lesions in the rabbit. Arthroscopy 1996, 12, 280-286.
  24. Madry, H.; Kon, E.; Condello, V.; Peretti, G.M.; Steinwachs, M.; Seil, R.; Berruto, M.; Engebretsen, L.; Filardo, G.; Angele, P. Early osteoarthritis of the knee. Knee Surg Sports Traumatol Arthrosc 2016, 24, 1753-1762.

Reviewer 2 Report

The study provides useful insight into the changes in biochemical composition of repair cartilage in in vivo sheep models. Biochemical composition measured include total DNA, total proteoglycans (PG), PG/DNA, total Col II, Col II/DNA, total Col I, Col I/DNA, Col I/Col II. The study was primarily focused on answering two questions and below were the main findings:

  • Comparison of biochemical composition of the repair tissue formed at the 4 x 8mm full-thickness chondral focal defect site in the lateral femoral trochlea of adult merino sheep (n=13) after 6 mo between normal cartilage removed as defect (t=0) vs 1 mm drill holes (n=7, t=6 mo) vs 1.8 mm drill holes (n=6, t=6mo) vs adjacent cartilage (t= 6mo)
    1. Normal cartilage (1.0 vs 1.8mm at t =0) -> no significant difference; repair cartilage (1.0 vs 1.8mm at t =6 mo) -> no significant difference ; adjacent cartilage (1.0 vs 1.8mm at t =6 mo) -> no significant difference a
    2. Normal (1 mm, t=0) vs repair cartilage (1 mm, t=6mo) -> significant differences were seen wrt total DNA, total PG, total Col I, Col I/DNA ; Normal (1mm, t=0) vs adjacent cartilage (1mm, t=6mo) -> significant differences were seen wrt total DNA, total PG, total Col I, Col I/DNA and Col I/Col II
    3. Normal (1.8 mm, t=0) vs repair cartilage (1.8 mm, t=6mo) -> significant differences were seen wrt total Col I, Col I/DNA, Col I/Col II ; Normal (1.8 mm, t=0) vs adjacent cartilage (1.8 mm, t=6mo) -> significant differences were seen wrt total DNA, total PG, total Col I, Col I/DNA and Col I/Col II
    4.  
  • Correlation of biochemical composition vs previously published (by the same authors) cartilage histology and micro-CT of SCB repair

Major concerns:

  1. The study has small number of animals in each group of repair cartilage assessment to make assertive conclusions. In addition, some of the conclusions were not validated with data or were incorrect:
    1. First conclusion – “The first major finding is that only 1.0 mm diameter subchondral drilling 30 significantly increases both DNA and proteoglycan contents of the cartilaginous repair tissue 31 compared to the original cartilage, but not 1.8 mm drilling, supporting the value of small diameter 32 bone cutting devices for marrow stimulation”.  In the 1.0mm repair group, there were 2 animals that show increased DNA and PG content (fig 1a,c), the performance of the remaining animals between the two groups (1.0mm vs 1.8mm) seem to be comparable.
    2. Second conclusion – “DNA contents correlated with the proteoglycan 33 and type-II collagen contents (which also correlated among themselves)”, no statistical analysis is provided for this conclusion?
    3. Third conclusion – “signs of early tissue degeneration were already present both within the (fibro)cartilaginous repair tissue, as indicated by the higher biochemical indices for type-I collagen, and the articular cartilage adjacent to the defects, as revealed by the significantly increased DNA, proteoglycan, and type-I collagen contents without differences between the two treatment groups at 6 months in vivo.” This conclusion seems to be incorrect. It is possible that due to the focal defect created, the adjacent cartilage tissue might also show OA progression biochemical changes? Significant differences were seen between normal cartilage vs repair cartilage and normal cartilage vs adjacent cartilage.

Minor comments:

  1. In introduction, line 2 – would be useful to have a quantitative definition for “small articular defects”.
  2. In introduction, line 4, it should be multipotent progenitor cells and not “pluripotent progenitor cells”
  3. Introduction - Page2, Line 2, please quantitatively define “small versus large diameter perforations”
  4. Readability needs to be improved throughout the manuscript, eg. “Besides, the biochemical composition of the repair cartilage with those of the original articular cartilage and with the cartilage adjacent to the defects has been studied only very infrequently [10,11]”
  5. In the methods section, Incorrect to state “Cartilage defects were treated by subchondral drilling applying two different hole diameters (1.0 and 1.8 mm)”, cartilage defect was 4 x 8mm rectangular full-thickness cartilage removal
  6. Instead of “original cartilage” group, probably using “normal cartilage” would be clearer

Author Response

Point-by-point response to the Reviewers’ comments

We were delighted to receive the review for our manuscript, and appreciate the Reviewers’ thorough reviews and insightful comments and suggestions to improve the quality and impact of our work. We have addressed each of the concerns below, on a point-by-point basis and making changes to the manuscript where appropriate. We believe that the manuscript has been significantly strengthened as a result of these recommendations. We very much appreciate the opportunity to submit this revised manuscript, and do hope that, with these changes and clarifications, our work might be suitable for publication in the Journal of Clinical Medicine.

Thanks again and best regards,

Henning Madry

Reviewer 2

1. Reviewer comment: The study provides useful insight into the changes in biochemical composition of repair cartilage in in vivo sheep models.

Author response: We thank Reviewer 1 very much for the positive evaluation and the very constructive and helpful review of our manuscript.  

Author action: None.

2. Reviewer comment: The study has small number of animals in each group of repair cartilage assessment to make assertive conclusions.

Author response: We again thank the Reviewer very much for this thoughtful remark. In the present investigation, we included a total number of n = 13 adult Merino sheep, i.e. 6-7 experimental large animals per group.

For the earlier published histological and micro-CT data in these same animals (termed historical control throughout the manuscript) [5], we had previously calculated sample size requirements based on data from other translational studies in cartilage defect models [10-13]. To detect a mean difference of 6 in total points for the histological scoring, a total of 6 animals would have been required when a unilateral surgical approach was used. These calculations were based on the assumption of a standard deviation of 3 points [14]. To accommodate for the possible loss of 1 animal due to complications, a sample size of 7 animals was chosen in this first investigation and also adopted for the here presented investigation. Eldracher et al. [5] demonstrated that the chosen group sizes allowed for the detection of significant differences between treatment groups, suggesting that the study was sufficiently powered because the magnitudes of changes in the histological and micro-CT analysis were comparably large. Besides, the number of experimental animals selected was sufficient to also reveal statistically significant differences in biochemical parameters between normal and repair and adjacent articular cartilage and may thus be regarded to be well within the range for the detection of significant effect sizes [14].

In fact, a literature review could demonstrate that in the scientific field of osteochondral repair, the majority of published investigations applied less than 13 large animals [8]. Thus, the here reported number of animals may be regarded to be adequate also when compared with the existing literature.

Author action: We have added a more detailed explanation of the sample size calculation to the revised Methods section in the new version of our manuscript (page 7, lines 7-10) and also have amended the resepctive reference of Orth et al. [14]; the reference list has been renumbered accordingly (page 32, reference #18).

3. Reviewer comment: In addition, some of the conclusions were not validated with data or were incorrect: First conclusion – “The first major finding is that only 1.0 mm diameter subchondral drilling significantly increases both DNA and proteoglycan contents of the cartilaginous repair tissue compared to the original cartilage, but not 1.8 mm drilling, supporting the value of small diameter bone cutting devices for marrow stimulation”.  In the 1.0mm repair group, there were 2 animals that show increased DNA and PG content (fig 1a,c), the performance of the remaining animals between the two groups (1.0mm vs 1.8mm) seem to be comparable.

Author response: We thank the Reviewer very much for this attentive remark. Instead of simply presenting the average (or mean) value for all individual animals in their respective groups, followed by a comparison of these groups as in any standard statistical analysis, we here chose to specifically visualize all these individual values in Figures 1 and 2. The aim of such graphical presentation was to attract the readers’ attention to the trends of these values per animal (in addition to their mean value which is always indicated as dotted red line in each image of Figure 1 and 2). Indeed, although few animals showed not always an increase for every single biochemical parameter (e.g. Figure 1 A: 0x, Figure 1 B: 2x, Figure 1 C: 1x, Figure 1 D: 1x), the statistical analysis of the means revealed that only 1.0 mm diameter subchondral drilling significantly increases both DNA and proteoglycan contents. This was not found for the mean values of DNA and proteoglycan contents following 1.8 mm drilling and brought us to the conclusion given above. Figure 1 (A-D) also demonstrates that in the 1.8 mm group 2 animals exerted a negative trend with decreased DNA values (Figure 1 B) and 1 animal presented with decreased proteoglycan values (Figure 1 D) in the repair cartilage compared to the normal cartilage. Such a decrease in these biochemical parameters was not found in any animal of the 1.0 mm drilling group (only 1 animal presented with unchanged values for proteoglycans normal versus repair tissue; Figure 1 C). To put these data and significances into a meaningful and clinically relevant perspective, we believe that the presentation of x-fold differences between normal articular cartilage (as gold standard) and the respective individual cartilage repair tissue as well as the adjacent cartilage is of interest.

Author action: In order to make these findings more accessible to the readership of the Journal of Clinical Medicine, x-fold differences between normal articular cartilage and the adjacent cartilage had already been given in the initially submitted version of the manuscript (pages 11 and 12). In addition, we now add such x-fold differences for the comparison between normal cartilage and the cartilaginous repair tissue (page 11, lines 5, 9, and 10). In order to make clear that red dotted lines in Figure 1 and 2 represent the mean values, an additional explanation is now added to the legends of both figures (page 16, line 10; page 18, lines 10-11). We also have added to the Discussion section that specifically mean values for DNA and proteoglycans have been significantly increased following 1.0 mm drilling (page 22, lines 10-14) and acknowledge the fact that not all individual animals always exhibited such numerical improvements (page 24, lines 4-9). 

4. Reviewer comment: Second conclusion – “DNA contents correlated with the proteoglycan and type-II collagen contents (which also correlated among themselves)”, no statistical analysis is provided for this conclusion?

Author response: We again thank the Reviewer for mentioning the crucial point of correlation analysis. We would like to point out most politely that in the initially submitted version of the manuscript, the results of the correlation analysis within the cartilaginous repair tissue were in its entirety presented on page 12 while the respective statistical methodology was explained on pages 8 and 9.

Author action: The data of the correlation analysis between DNA, proteoglycan and type-II collagen contents of the repair tissue are presented on page XXX(12) in the revised version of our manuscript, reading as follows: „In both treatment groups, DNA contents of the repair tissue correlated with the proteoglycan contents (r ≥ 0.829; P ≤ 0.042) and with its type-II collagen contents (r ≥ 0.714; P ≤ 0.005). The type-II collagen contents also correlated with the proteoglycan contents (r ≥ 0.943; P ≤ 0.005). Other correlations were beyond statistical significance (data not shown).“

Furthermore, results of the correlation analysis (1) among biochemical parameters within the adjacent articular cartilage and (2) between biochemical parameters of the repair tissue and the historical histological and micro-CT parameters of the repaired osteochondral unit (obtained from [5]) are also given on pages 12 and 13 of the revised manuscript.  

5. Reviewer comment: Third conclusion – “signs of early tissue degeneration were already present both within the (fibro)cartilaginous repair tissue, as indicated by the higher biochemical indices for type-I collagen, and the articular cartilage adjacent to the defects, as revealed by the significantly increased DNA, proteoglycan, and type-I collagen contents without differences between the two treatment groups at 6 months in vivo.” This conclusion seems to be incorrect. It is possible that due to the focal defect created, the adjacent cartilage tissue might also show OA progression biochemical changes? Significant differences were seen between normal cartilage vs repair cartilage and normal cartilage vs adjacent cartilage.

Author response: We thank the Reviewer for this very valuable comment and we absolutely agree that the focal articular cartilage defects created in this animal model induce early osteoarthritis in the adjacent cartilaginous tissue. Our results showed degenerative changes within the adjacent articular cartilage following either subchondral drilling technique, exhibiting an up to 55-fold increase in type-I collagen content compared with the normal cartilage. Besides, both DNA and proteoglycan contents were significantly elevated (2.6- and 2.2-fold, respectively), most likely representing an early attempt to restore extracellular matrix components adjacent to the lesion as regularly observed during the early hypertrophic phase of osteoarthritis, suggesting that at this time point the rate of repair is greater than the rate of degradation [15]. Thus, in summary these findings point towards an early but not advanced stage of osteoarthritis within the articular cartilage adjacent to the treated defects [16]. Similar findings have already been described by other groups for both translational large animal models such as goats of experimentally induced critical-sized cartilage defcets [17] as well as patients suffering from osteochondrosis dissecans and surgical fragment excision in a prolonged clinical investigations at 16 years postoperatively [18].

To answer the fine point of the Reviewer whether both therapeutic drilling approaches (1.0 and 1.8 mm) in fact have the potential to reduce such early degenerative changes within the repair tissue as well as within the adjacent cartilage, a control group of identically created cartilage defects but without any further treatment by subchondral drilling (i.e. untreated cartilage removal sites; please see also comment #10 of this Reviewer) would be necessary. For the here included animals, this is not feasible post mortem. However, we have addressed this interesting topic in another translational investigation applying the microfracture technique instead of subchondral drilling for marrow stimulation of full-thickness cartilage defects in 16 adult Merino sheep [12]. We demonstrated that neither of two different microfracture awls (1.0 and 1.2 mm in diameter) was able to prevent signs of early osteoarthritis within the adjacent cartilage surrounding the defect sites compared with solely debrided (i.e. untreated) control defects. For reasons of duplicate publications, we would however prefer not use these historical data in the methods section of the current investigation.

Author action: In order to address the highly interesting question whether subchondral drilling per se has the potential to reduce or even prevent signs of early osteoarthritis in the repaired and adjacent articular cartilage, we now have amended the Discussion section of the revised manuscript accordingly. The need for future investigations on this specific aspect raised by the Reviewer is pointed out more clearly (page 25, line 25 through page 25, line 7).    

6. Reviewer comment: In introduction, line 2 – would be useful to have a quantitative definition for “small articular defects”

Author response: We again thank the Reviewer for this comment. In the clinical setting, articular cartilage defect below 3 cm2 (300 mm2) are the classical indication for marrow stimulation procedures such as subchondral drilling [19].

As the defects created in the present study were comparably small (32 mm2; ovine

medial condylar width: 19 mm [20]) compared with the human medial condylar width of 27 mm [21], the defects created here may thus be regarded as an adequate animal model of small articular cartilage defects/removal sites.

Author action: As requested by the Reviewer, we now have added the quantitative definition for “small articular defects” given above (< 3 cm2) to the Introduction section of the manuscript (page 4, lines 3-5).

7. Reviewer comment: In introduction, line 4, it should be multipotent progenitor cells and not “pluripotent progenitor cells”

Author response: We thank the Reviewer for this remark.

Author action: The term “pluripotent” has been changed to “multipotent” in the revised version of the manuscript (page 4, line 8).

8. Reviewer comment: Introduction - Page2, Line 2, please quantitatively define “small versus large diameter perforations”

Author response: We again thank the Reviewer very much for this crucial and thoughtful observation. Indeed, the definition of small and large diameter perforations is an interesting topic comprising two different aspects: First the absolute diameter of the instrument applied for subchondral bone perforation, second the relationship between perforation and the entire defect area.

Regarding the first aspect of absolute instrument diameter, a systematic literature analysis [1] shows that for subchondral drilling procedures in animal models, only very few comparative data are available. In general, the average diameter of the drilling device was of 1.1 ± 0.5 mm. In small preclinical models, the smallest instrument (0.2 mm) was used in rats [22] while the largest (2.0 mm) was applied in rabbits [23]. In large preclinical models of subchondral drilling, only the previously cited study of Eldracher et al. [5] is available which also served as historical control for the present work, applying 1.0 and 1.8 mm diameter K-wires.

Regarding the second aspect of relative instrument diameter, the mean ratio between the entire defect area and the perforation area is of 7.7 ± 4.4 in preclinical animal models of subchondral drilling and of 5.6 in large animal models [1].

In the present investigation, we applied K-wires of 1.0 mm and 1.8 mm in diameter to create a total of 6 drill holes within a cartilage removal site of 32 mm2 (rectangular; 4 x 8 mm). Therefore, the absolute dimensions of the here created perforations range between the existing maximum and minimum literature values for this translational model [5]. Further, these perforations lead to average area ratios of 2.1 (1.0 mm group) and 6.8 (1.8 mm group).

Taking these calculations and literature data in large animal models into consideration, absolute instrument diameters of ≤ 1.0 mm and resulting area ratios ≤ 3 may be regarded as “small”. Absolute instrument diameters ≥ 1.5 mm or area ratios > 3 may be regarded as “large”.    

Author action: We now have added the above calculated absolute threshold values for the definition of „small versus large diameter perforations“ to the respective text passage in the Introduction section (page 4, lines 18-21).     

9. Reviewer comment: Readability needs to be improved throughout the manuscript, eg. “Besides, the biochemical composition of the repair cartilage with those of the original articular cartilage and with the cartilage adjacent to the defects has been studied only very infrequently [10,11]”

Author response: We thank the Reviewer for this remark and acknowledge that readability of this sentence was poor.

Author action: We now have rephrased the respective text passage. It now reads “Besides, the biochemical composition of the repair cartilage has been compared only very infrequently with those of the original articular cartilage and the cartilage adjacent to the defects [10,11] and nearly never in the context of marrow stimulation.” (page 4, line 25 through page 5, line 2)

10. Reviewer comment: In the methods section, Incorrect to state “Cartilage defects were treated by subchondral drilling applying two different hole diameters (1.0 and 1.8 mm)”, cartilage defect was 4 x 8mm rectangular full-thickness cartilage removal

Author response: We thank the Reviewer very much for this thoughtful remark. The term “cartilage defect” has been chosen according to the established terminology in the current literature. However, we fully agree that we have created and treated a rectangular area of full-thickness articular cartilage removal (also termed debridement). This artificially created cartilage removal site has been immediately treated by subchondral drilling in a one-step approach (i.e. cartilage removal and surgical treatment at the same time). Of course this methodological approach only partially reflects the clinical situation in which non-acute/chronic and sometimes even degenerative chondral lesions are much more common [24]. Nevertheless and considering economic feasibility and practicality as well as the restrictive regulations in animal experimentations, this study design is generally widely accepted in the field of osteochondral repair; treatment evaluation of chronic defects has been performed only very seldom [9]. 

Author action: The term „cartilage defect“ has been changed to „cartilage removal site“ in numerous sentences throughout the manuscript (page 6, line 9; page 6, line 12; page 7, line 16; page 7, line 21; page 25, line 25; page 26, lines 3-4; page 27, lines 5-6). In order to facilitate readability and text flow, we suggest to keep the term „defect“ at several text passages where appropriate.

11. Reviewer comment: Instead of “original cartilage” group, probably using “normal cartilage” would be clearer

Author response: We thank the Reviewer very much for this excellent suggestion.

Author action: At several passages throughout the revised version of the manuscript, we have changed the term „original cartilage“ to „normal cartilage“ (page 6, line 11; page 6, line 19; page 8, line 3; page 10, line 5; page 11, line 4; page 11, line 21; page 24, line 19; page 25, line 16).

References

  1. Gao, L.; Goebel, L.K.H.; Orth, P.; Cucchiarini, M.; Madry, H. Subchondral drilling for articular cartilage repair: a systematic review of translational research. Dis Model Mech 2018, 11.
  2. Chen, H.; Chevrier, A.; Hoemann, C.D.; Sun, J.; Lascau-Coman, V.; Buschmann, M.D. Bone marrow stimulation induces greater chondrogenesis in trochlear vs condylar cartilage defects in skeletally mature rabbits. Osteoarthritis Cartilage 2013, 21, 999-1007.
  3. Orth, P.; Cucchiarini, M.; Kaul, G.; Ong, M.F.; Graber, S.; Kohn, D.M.; Madry, H. Temporal and spatial migration pattern of the subchondral bone plate in a rabbit osteochondral defect model. Osteoarthritis Cartilage 2012, 20, 1161-1169.
  4. Chen, H.; Chevrier, A.; Hoemann, C.D.; Sun, J.; Ouyang, W.; Buschmann, M.D. Characterization of subchondral bone repair for marrow-stimulated chondral defects and its relationship to articular cartilage resurfacing. Am J Sports Med 2011, 39, 1731-1740.
  5. Eldracher, M.; Orth, P.; Cucchiarini, M.; Pape, D.; Madry, H. Small subchondral drill holes improve marrow stimulation of articular cartilage defects. Am J Sports Med 2014, 42, 2741-2750.
  6. Orth, P.; Cucchiarini, M.; Wagenpfeil, S.; Menger, M.D.; Madry, H. PTH [1-34]-induced alterations of the subchondral bone provoke early osteoarthritis. Osteoarthritis Cartilage 2014, 22, 813-821.
  7. Orth, P.; Cucchiarini, M.; Zurakowski, D.; Menger, M.D.; Kohn, D.M.; Madry, H. Parathyroid hormone [1-34] improves articular cartilage surface architecture and integration and subchondral bone reconstitution in osteochondral defects in vivo. Osteoarthritis Cartilage 2013, 21, 614-624.
  8. Orth, P.; Cucchiarini, M.; Kohn, D.; Madry, H. Alterations of the subchondral bone in osteochondral repair--translational data and clinical evidence. Eur Cell Mater 2013, 25, 299-316.
  9. Lind, M.; Larsen, A.; Clausen, C.; Osther, K.; Everland, H. Cartilage repair with chondrocytes in fibrin hydrogel and MPEG polylactide scaffold: an in vivo study in goats. Knee Surg Sports Traumatol Arthrosc 2008, 16, 690-698.
  10. Goebel, L.; Orth, P.; Muller, A.; Zurakowski, D.; Bucker, A.; Cucchiarini, M.; Pape, D.; Madry, H. Experimental scoring systems for macroscopic articular cartilage repair correlate with the MOCART score assessed by a high-field MRI at 9.4 Tesla - comparative evaluation of five macroscopic scoring systems in a large animal cartilage defect model. Osteoarthritis Cartilage 2012, 20, 1046-1055.
  11. Goebel, L.; Zurakowski, D.; Muller, A.; Pape, D.; Cucchiarini, M.; Madry, H. 2D and 3D MOCART scoring systems assessed by 9.4 T high-field MRI correlate with elementary and complex histological scoring systems in a translational model of osteochondral repair. Osteoarthritis Cartilage 2014, 22, 1386-1395.
  12. Orth, P.; Duffner, J.; Zurakowski, D.; Cucchiarini, M.; Madry, H. Small-Diameter Awls Improve Articular Cartilage Repair After Microfracture Treatment in a Translational Animal Model. Am J Sports Med 2016, 44, 209-219.
  13. Orth, P.; Goebel, L.; Wolfram, U.; Ong, M.F.; Graber, S.; Kohn, D.; Cucchiarini, M.; Ignatius, A.; Pape, D.; Madry, H. Effect of subchondral drilling on the microarchitecture of subchondral bone: analysis in a large animal model at 6 months. Am J Sports Med 2012, 40, 828-836.
  14. Orth, P.; Zurakowski, D.; Alini, M.; Cucchiarini, M.; Madry, H. Reduction of sample size requirements by bilateral versus unilateral research designs in animal models for cartilage tissue engineering. Tissue Eng Part C Methods 2013, 19, 885-891.
  15. Squires, G.R.; Okouneff, S.; Ionescu, M.; Poole, A.R. The pathobiology of focal lesion development in aging human articular cartilage and molecular matrix changes characteristic of osteoarthritis. Arthritis Rheum 2003, 48, 1261-1270.
  16. Madry, H.; Luyten, F.P.; Facchini, A. Biological aspects of early osteoarthritis. Knee Surg Sports Traumatol Arthrosc 2012, 20, 407-422.
  17. Schinhan, M.; Gruber, M.; Vavken, P.; Dorotka, R.; Samouh, L.; Chiari, C.; Gruebl-Barabas, R.; Nehrer, S. Critical-size defect induces unicompartmental osteoarthritis in a stable ovine knee. J Orthop Res 2012, 30, 214-220.
  18. Sanders, T.L.; Pareek, A.; Obey, M.R.; Johnson, N.R.; Carey, J.L.; Stuart, M.J.; Krych, A.J. High Rate of Osteoarthritis After Osteochondritis Dissecans Fragment Excision Compared With Surgical Restoration at a Mean 16-Year Follow-up. Am J Sports Med 2017, 45, 1799-1805.
  19. Gomoll, A.H.; Farr, J.; Gillogly, S.D.; Kercher, J.; Minas, T. Surgical management of articular cartilage defects of the knee. J Bone Joint Surg Am 2010, 92, 2470-2490.
  20. Osterhoff, G.; Loffler, S.; Steinke, H.; Feja, C.; Josten, C.; Hepp, P. Comparative anatomical measurements of osseous structures in the ovine and human knee. Knee 2011, 18, 98-103.
  21. Murshed, K.A.; Cicekcibasi, A.E.; Karabacakoglu, A.; Seker, M.; Ziylan, T. Distal femur morphometry: a gender and bilateral comparative study using magnetic resonance imaging. Surg Radiol Anat 2005, 27, 108-112.
  22. Hamanishi, M.; Nakasa, T.; Kamei, N.; Kazusa, H.; Kamei, G.; Ochi, M. Treatment of cartilage defects by subchondral drilling combined with covering with atelocollagen membrane induces osteogenesis in a rat model. J Orthop Sci 2013, 18, 627-635.
  23. Menche, D.S.; Frenkel, S.R.; Blair, B.; Watnik, N.F.; Toolan, B.C.; Yaghoubian, R.S.; Pitman, M.I. A comparison of abrasion burr arthroplasty and subchondral drilling in the treatment of full-thickness cartilage lesions in the rabbit. Arthroscopy 1996, 12, 280-286.
  24. Madry, H.; Kon, E.; Condello, V.; Peretti, G.M.; Steinwachs, M.; Seil, R.; Berruto, M.; Engebretsen, L.; Filardo, G.; Angele, P. Early osteoarthritis of the knee. Knee Surg Sports Traumatol Arthrosc 2016, 24, 1753-1762.
